# Potential of Impedance Flow Cytometry to Assess the Viability and Quantity of *Cannabis sativa* L. Pollen

**DOI:** 10.3390/plants10122739

**Published:** 2021-12-13

**Authors:** Hamza Rafiq, Jens Hartung, Lisa Burgel, Georg Röll, Simone Graeff-Hönninger

**Affiliations:** 1Department of Agronomy, Institute of Crop Science, Cropping Systems and Modelling, University of Hohenheim, 70599 Stuttgart, Germany; lisa.burgel@uni-hohenheim.de; 2Department of Agronomy, Institute of Crop Science, Biostatistics, University of Hohenheim, 70599 Stuttgart, Germany; jens.hartung@uni-hohenheim.de; 3Amphasys AG, Technopark Lucerne, 6039 Root, Switzerland; georg.roell@amphasys.com

**Keywords:** gibberellic acid, male flower, silver thiosulfate solution, total number of pollen cells

## Abstract

Over the last decade, efforts to breed new *Cannabis sativa* L. cultivars with high Cannabidiol (CBD) and other non-psychoactive cannabinoids with low tetrahydrocannabinol (THC) levels have increased. In this context, the identification of the viability and quantity of pollen, which represents the fitness of male gametophytes, to accomplish successful pollination is of high importance. The present study aims to evaluate the potential of impedance flow cytometry (IFC) for the assessment of pollen viability (PV) and total number of pollen cells (TPC) in two phytocannabinoid-rich cannabis genotypes, KANADA (KAN) and A4 treated with two different chemical solutions, silver thiosulfate solution (STS) and gibberellic acid (GA3). Pollen was collected over a period of 8 to 24 days after flowering (DAF) in a greenhouse experiment. Impedance flow cytometry (IFC) technology was used with *Cannabis sativa* to assess the viability and quantity of pollen. The results showed that the number of flowers per plant was highest at 24 DAF for both genotypes, A4 (317.78) and KAN (189.74). TPC induced by STS was significantly higher compared to GA3 over the collection period of 8 to 24 DAF with the highest mean TPC of 1.54 × 10^5^ at 14 DAF. STS showed significantly higher viability of pollen compared to GA3 in genotype KAN, with the highest PV of 78.18% 11 DAF. Genotype A4 also showed significantly higher PV with STS at 8 (45.66%), 14 (77.88%), 18 (79.37%), and 24 (51.92%) DAF compared to GA3. Furthermore, counting the numbers of flowers did not provide insights into the quality and quantity of pollen; the results showed that PV was highest at 18 DAF with A4; however, the number of flowers per plant was 150.33 at 18 DAF and was thus not the maximum of produced flowers within the experiment. IFC technology successfully estimated the TPC and differentiated between viable and non-viable cells over a period of 8 to 24 DAF in tested genotypes of *Cannabis sativa*. IFC seems to be an efficient and reliable method to estimate PV, opening new chances for plant breeding and plant production processes in cannabis.

## 1. Introduction

The global demand for *Cannabis sativa* L. has been growing rapidly in various branches of industry (e.g., pharmacy, food, and cosmetics) for several years. In parallel, new cultivars are being bred and are entering the market, catching up with the different sectoral needs. Breeding efforts on phytocannabinoid-rich genotypes with a high content of cannabidiol (CBD), cannabigerol (CBG), cannabichromene (CBC), etc. aim to keep the tetrahydrocannabinol (THC) level below 0.2%, as currently mandated by the European Union (EU) [1,2]. These phytocannabinoid-rich:low THC cultivars are still rarely available in the EU and have to be obtained through targeted breeding programs [3]; in contrast, the THC dominant cultivars are comparatively available on a broad range due to the former artificial human selection for their psychoactive and medicinal properties [4].

Breeding in cannabis for phytocannabinoid-rich genotypes is challenging as cannabis is a dioecious plant. Thus, its heterogametic male (XY) and homogametic female (XX) inflorescences appear on separate plants [5]. Commercially, female plants are produced on a large scale and have been given more importance based on their secretory hairs, known as glandular trichomes, that cover the inflorescences. Female flowers synthetize phytocannabinoids, terpenes, and other secondary metabolites, posing medicinal applications for several therapeutic applications [6,7]. Male flowers, which predominantly produce pollen, usually have a very low cannabinoid concentration [8].

Sex expression becomes evident during generative growth stage. The late appearance of male flowers (XY) in a female plant production facility poses a risk of pollination. Regardless of its dioecious nature, hermaphroditism or staminate inflorescences produced on female dominant plants (XX) can be triggered by environmental stresses such as prolonged periods of darkness during the early flowering stage, inadequate nutrition, and low temperature [9,10]. In addition, hermaphrodites on predominantly female plants can be deliberately induced by different chemical applications, such as silver nitrate (AgNO_3_), silver thiosulfate solution (STS), and gibberellic acid (GA3) [11,12]. In *Cannabis sativa*, previous studies [12,13,14] have reported that STS and GA3 act as a ethylene antagonists and promote the production of male flowers on female plants. Pollen produced through hermaphrodite plants are feminized in nature, which produces feminized seeds after pollination, as both parents share their X chromosome [12,15].

The fitness of produced pollen could be efficiently and quickly evaluated through viability testing [16,17]. Viability of pollen in general is highly vulnerable to environmental changes during flowering, such as temperature and humidity [18]. A high amount of pollen produced per flower does not guarantee successful pollination unless the viability of pollen is secured by harvesting it at an optimum time in compatible parents [19]. Various histochemical methods have been used to distinguish alive and aborted pollen, such as the tetrazolium salts and Alexander’s stain method [20,21]. Other non-cytotoxic staining methods use fluorescein diacetate (FDA) in cell permeable membranes to analyze the pollen viability (PV), but with the limitation of false positive results [22,23]. Pollen tube germination is another method to determine PV based on the ability of pollen tube formation. This method is highly dependent on the germination media and the environmental conditions [21]. Reports of line dependency on the germination buffer were also reported by Heidmann et al. [24] and Rodriguez-Riano and Dafni [25]. These microscopic methods require trained, experienced scientists and can take up to several hours to obtain the result. An accurate and rapid method is necessary to overcome these obstacles in the determination of PV, as well as their quantity by collecting the pollen at various times of flowering.

Impedance flow cytometry (IFC) technology has demonstrated to be a fast, reliable, and label-free technology to study the pollen quantity and viability simultaneously [24]. IFC characterizes dead and viable pollen cells in a broad range of species such as wheat, corn, hazelnut, tomato, tobacco, etc. [26,27]. In contrast to traditional methods, IFC determines the electrical properties of cells by using microfluidic chips and analyzes a high throughput of pollen cells for viability screening in a standardized method, up to 1000 cells per second [26]. IFC can be used in the field or the greenhouse to analyze pollen on a large scale during the screening of breeding lines [24]. To our knowledge, IFC has not been used in cannabis so far.

The aim of this study was to evaluate the potential of IFC technology to assess (i) PV and (ii) TPC in two cannabis genotypes KANADA (KAN) and A4. The two chemical solutions STS and GA3 were investigated in terms of their impact on PV and TPC of induced male flowers on female dominant plants of the two cannabis genotypes. In addition, pollen was analyzed at different days after flowering (DAF) to study the impact of DAF on PV and TPC within and between genotypes.

## 2. Results and Discussion

### 2.1. Number of Flowers Per Plant

All plants treated with GA3 and STS solutions produced male flowers in this experiment. The number of flowers per plant showed significant interactions between solutions and DAF (Table 1). Plants treated with STS produced a maximum number of flowers at 24 DAF (273.33). The number of flowers per plant produced by STS at 24 DAF were significantly higher when compared to sampling dates 11 (37.14), 14 (157.02), and 18 (174.30) DAF (Figure 1a). For both solutions, the number of flowers per plant increased with time. While the final number of flowers was similar, GA3 had lower numbers at early sampling dates (Figure 1a). Plants treated with GA3 also produced the highest number of flowers per plant (220.60) 24 DAF. Thus, both solutions resulted in a higher number of flowers per plant at later DAF.

Furthermore, the interaction between genotype and DAF was significant for the number of flowers per plant (Table 1). The genotype KAN produced the highest number of flowers (189.74) 24 DAF. The number of flowers produced by KAN at 24 DAF was significantly higher compared to 11 (7.95), 14 (94.45), and 18 (115.89) DAF. The genotype A4 showed a similar trend as number of flowers increased with later sampling dates. The maximum flower production (317.78) was observed 24 DAF. Nevertheless, genotype A4 had a significantly higher number of flowers at sampling dates of 11, 21, and 24 DAF compared to KAN (Figure 1b).

Ethylene hormone has been shown to enhance the female sex expression [28]; other studies [12,13,14] have reported that STS and GA3 promote male flowering by competing with ethylene hormone. In a recent study, Adal et al. [29] identified 200 genes that could potentially control male sex expression on female dominant plants of cannabis; however, the authors suggested further investigation of these genes through overexpression and knockout experiments required to better understand the opposite sex flower production in cannabis plants.

The results (Figure 1) indicated that the number of flowers per plant increased with time for the selected chemical solution and genotype. Ram et al. [11] compared different types of gibberellins (GA3, GA7, GA9, and GA4 + 7) at various concentrations of 25, 50, 75, and 100 μg to induce male flowers in genetically female plants of *Cannabis sativa*. They reported that male flowers were stimulated with all forms of gibberellins, but GA3 with 50 μg led to a maximum (35.0) of flowers per plant. Lubell et al. [14] treated four cultivars of female hemp with 0.3 and 3 mM STS solution and reported 100% conversion to male flowers at 3 mM concentration. Differences in the number of flowers between STS and GA3 were observed by Flajšman et al. [30], who reported 379 total male flowers with 20 mM STS and below 25 male flowers treated with 0.01% GA3 on two CBD-rich breeding populations. In our study, the number of male flowers produced between 21 and 24 DAF were similar for both solutions; however, TPC determined by IFC (Figure 2) showed that STS produced a significantly higher amount of pollen compared to GA3 at all sampling dates.

### 2.2. Effect of Chemical Solution on TPC at Various Sampling Dates

Plants treated with STS produced flowers with the highest mean TPC of 1.54 × 10^5^ at 14 DAF, and lowest at 24 DAF with a mean TPC of 6.67 × 10^4^. Solution STS showed an increasing trend of TPC between 8 and 14 DAF, followed by a decrease in the mean value of TPC between 14 and 24 DAF (Figure 2)**,** while the number of flowers increased with sampling dates between 14 and 24 DAF (Figure 1).

Plants treated with STS produced flowers with TPC of approximately 100 times higher compared to GA3. The mean TPC produced by GA3 ranged between 850 and 1500 TPC. Significant differences in the mean TPC of flowers were found between the two solutions on each sampling date over the period of 8 to 24 DAF. In addition, no significant differences were found between the two tested genotypes in terms of produced TPC.

The pollen grain quantity might be one of the key traits in pollinators parents to improve hybrid seed production [31,32]. Ram et al. [12] and Rosenthal [13] reported that STS and GA3 with specific concentration and treatment duration have the tendency to promote the development of male flowers on female plants (XX) of cannabis. The results of this study confirmed the observation of Ram et al. [12], Rosenthal [13], and Green [33], as male flowers were produced for both genotypes, KAN and A4, by applying STS and GA3. In previous studies [12,14,30], the quantity of pollen was not considered while defining the efficiency of treatment to induced masculinity in female plants of cannabis. Our results reported that TPC were significantly different between two treatments at all sampling dates, while the difference in the number of flowers per plant was not significant between solutions and genotypes between 21 and 24 DAF (Figure 1a). According to Anfinrud [34], if male parental lines produced a limited number of pollen, the number of male plants should be increased compared to female plants to assure adequate amounts of seed stock. Based on our results, STS treatment produced significantly higher amounts of pollen cells (highest mean TPC of 1.54 × 10^5^ at 14 DAF) compared to GA3 on all sampling dates. Therefore, the number of male plants can be reduced in a breeding scheme of *Cannabis sativa*, as sufficient amounts of pollen grains would be available for pollination [10]. The pollen sampling window of 14 to 18 DAF seemed to be ideal to collect high numbers of pollen with STS treatment.

### 2.3. Effect of Chemical Solution on PV at Various Sampling Dates

The PV of flowers showed significant interactions between genotypes, solution type, and DAF (Table 1). While PV for GA3 compared to STS was lower on all DAF and both genotypes, the size of the differences varied. The largest values for PV were observed at intermediate DAF with a maximum for A4, STS on 18 DAF (79.36%). Genotype A4 with STS solution indicated that the PV of 18 DAF was not significantly different in comparison to sampling dates of 14 (77.88%), 21 (71.33%), and 24 (51.92%) DAF (Figure 3a). STS and GA3 at each sampling date of genotype A4 indicated that the viability of pollen when treated with STS was significantly different from GA3 at sampling dates of 8, 14, 18, and 21 DAF. Genotype A4 showed a lower PV value when treated with STS at the start of sampling date 8 (45.66%) and 11 (53.86%) DAF (Figure 3a).

The highest PV value (78.18%) was observed at 11 DAF in the genotype KAN under STS treatment (Figure 3b). The mean comparison of KAN indicated that PV at 11 DAF was not significantly different from 8 (71.23%), 14 (77.22%), 18 (67.20%), and 21 (70.43%) DAF. Mean comparison between solutions treatment at each sampling date indicated, for genotype KAN, a significantly higher viability of pollen when treated with STS in comparison to GA3 at sampling dates of 8, 11, 14, 18, and 21 DAF. PV of genotype KAN declined in the treatment STS 24 DAF (36.49%) but was not significantly different from GA3 at 24 DAF (14.88%). Treatment GA3 resulted in the highest PV of 34.90% at 11 DAF for genotype A4, while KAN produced maximum viability of 24.14% 14 DAF. Flowers of KAN treated with GA3 indicated the lowest viability of 0.63% at early harvest of 8 DAF.

Viability of pollen is sensitive to high temperatures and low humidity [16]. In this study, pollen was processed with IFC immediately after harvest to ensure maximum viability without exposing the pollen to environmental stress for a longer period. Finding an optimal time for pollen collection is critical for the viability of pollen [35]. Our study provided an overview of PV over a range of sampling dates between 8 and 24 DAF (Figure 3a) and indicated that sampling dates between 14 to 24 DAF can be recommended for genotype A4 treated with STS. Moreover, the sampling window for genotype KAN treated with STS was even larger compared to A4 and ranged from 8 to 21 DAF (Figure 3b), which seems to be ideal for early pollination. Our results showed that pollen could be harvested over an extended period of 17 days (8 to 24 DAF) (Figure 3a,b), which would allow flexibility of time to synchronize timing of female reproductive organs for successful pollination during cannabis breeding. Furthermore, the assessment of pollen quality and quantity in *Cannabis sativa* is crucial for the production of new cultivars and maintenance of biodiversity. To be competitive with imported material, the development of new cultivars in Europe is indispensable and needs new innovative techniques such as IFC that support the needed breeding processes.

## 3. Materials and Methods

### 3.1. Greenhouse Cultivation System

A greenhouse experiment was carried out to evaluate the potential of IFC by using two genotypes of *Cannabis sativa* L., KANADA (KAN) and A4 (obtained from AI FAME, Wald-Schönengrund, Switzerland). A control with no chemical solution was not added as the two genotypes produced only female flowers under this condition [36]. The experiment was conducted from 30 July 2020 until 2 November 2020, at the University of Hohenheim, Germany. The clonal production of the two genotypes was performed on 30 July 2020 by cutting the apical tips of mother plants [37]. A rooting gel (Clonex^®^, Lansing, MI, USA) consisting of the plant hormone (0.31% Indole-3-butyric acid) was applied for better development of root cells. Clonal cuts were grown in a 47 cm × 30 cm planting tray having growing media of 3.5 cm × 3.5 cm Eazy plugs^®^ (Goirle, the Netherlands). The planting tray was covered with a transparent hood and the plants were sprayed with water twice a day to keep the relative humidity around 90%. The range of temperature was maintained between 24 °C and 26 °C (day/night). Artificial lighting of CHD Agro 400 Watt (DH Licht GmbH, Wülfrath, Germany) was used to maintain an 18 h photoperiod. After 14 days, eight clonal cuts of each genotype were transplanted into round pots of 12 cm in diameter. A substrate composition of 80% Klasmann substrate 5 (590), consisting of 15% black peat, 20% fraction 1, 25% milled peat, 20% GF medium, 10% pine bark, 10% leca, 1 kg m^−3^ horn chips, and 1 kg m^−3^ NPK 12-14-24 (Klasmann-Deilmann GmbH, Geeste, Germany) was mixed with 20% perlite of Perligran^®^ Extra (KNAUF, Iphofen, Germany). Finally, plants were transplanted once again 28 days after plantation (DAP) into 4.5 L square pots (15 × 15 × 20 cm), resulting in a total of 16 plants with 4 replicates. These 16 plants (8 per genotype) were allocated in a four-by-four grid with two plants from each genotype in each row and column.

During the vegetative growth cycle, 18 h of light was provided in the greenhouse, supplementing sunlight with artificial light using CHD Agro 400 Watt. Plants were irrigated manually and fertilized four days a week with 0.2% Plantaactiv 18-12-18 Type A (Hauert, Grossaffoltern, Switzerland) during vegetative growth and with 0.2% Plantaactiv 10-20-30 Type B during generative stage following the protocol of Burgel et al. [38]. The light duration of the greenhouse chamber was shifted from 18 h to 12 h 41 DAP using black shaded curtains to block potential sunlight interference. The range of temperature was between 21.6 °C and 28.3 °C during the vegetative period and 20.6 °C and 28.6 °C during the generative growth cycle. The relative humidity varied from 40.6% to 67.3% during the vegetative period and 30.2% to 46.7% during the generative growth cycle.

### 3.2. Preparation of Chemical Solutions

STS was prepared by dissolving 0.1 g of silver nitrate (Carl Roth, Karlsruhe, Germany) in 100 mL distilled water and 0.5 g of sodium thiosulfate in 100 mL distilled water. The two solutions were finally mixed and diluted with 1:9 distilled water [33]. GA3 (Merck, Darmstadt, Germany) was prepared by dissolving 0.1 g of GA3 in 1 L of distilled water [13]. A standardized pruning technique was performed at the start of spraying, where lateral branches were removed to six internodes for both genotypes. A total of 48 DAP chemical solutions were evenly sprayed on all branches of the plants. Spraying was performed on every two consecutive days followed by two days without treatment until 69 DAP, when initiation of the first male flowers occurred, considered as the first day of flowering. Each solution was sprayed once on each genotype in each row and column. Thus, the experiment was finally designed as a Latin square design with four combinations of two genotypes and two solutions occurring in each row and column.

### 3.3. Pollen Viability Analysis with IFC

The pollen viability was analyzed by using the impedance flow cytometry (IFC) Ampha Z32 instrument (Amphasys AG, Root, Switzerland), which represents a fast and label free technology. After pollen extraction, they were suspended in an Ampha fluid 6 measurement buffer (Amphasys AG, Root, Switzerland). The pollen suspension was then pumped through a microfluidic chip. As dead and viable cells behave differently in an electric field, they can be detected and distinguished by the instrument. Results were graphically displayed as scatterplots in the AmphaSoft 2.0 Software (Amphasys AG, Root, Switzerland) (Figure 4).

To adjust the parameters to measure the cannabis pollen (size 25–30 μm), the instrument settings of a *Solanum lycopersicum* L. template (Trigger Level 0.1 V; Modulation 4; Amplification 6; Demodulation 2) were used. The Ampha fluid 6 (Amphasys AG, Root, Switzerland) measurement buffer was slightly modified by adding the detergent 0.05% Tween 20 to enable a homogeneous pollen suspension. The used microfluidic chip had the channel size of 120 μm specified by Amphasys as a D-chip. The cells were measured at frequencies: of 2 MHz and 18 MHz. For data evaluation, the cells were analyzed at 18 MHz by excluding air bubbles (hiding) and placing a vertical gate between the dead and viable population (Figure 4). A clear separation between dead and viable cells was always possible.

### 3.4. Sample Preparation and Measurements

Male flowers were produced on all plants of both genotypes. Each flower enclosed five anthers with outer layer sepals. During flower maturity, the anthers came out of the sepal but were still intact to stamen with attached filament. Individual flowers were carefully removed with forceps and collected in 1.5 mL Eppendorf tubes. To extract the pollen out of anther, the Eppendorf tube was scratched over the rack up to five times until the pollen was out of each anther. Five single flower samples were taken at each of six sampling dates (in a three-to-four-day interval) to Eppendorf tubes from each plant treated with STS and GA3. Additionally, a pooled sample of five flowers was taken from all 16 plants at all dates.

A measure of 1 mL of buffer solution was added to pollen in the Eppendorf tube and was filtered through a 100 μm filter to remove the residues. Finally, 2 mL of buffer solution was added to the tube to have 3 mL of final solution. The tube was put directly on the IFC for pollen analysis. The number of mature flowers was counted and removed (Figure 1) from each plant on sampling dates of 11, 14, 18, 21, and 24 DAF.

### 3.5. Statistical Analysis

Data were analyzed with a mixed model approach accounting for genotype, solutions, DAF, and the design. The model can be described as:$$y_{hijklmn}=\mu +r_{hm}+c_{im}+p_{him}+s_{j}+\tau_{k}+\varphi_{l}+\theta_{m}+{\left(\tau \varphi \right)}_{kl}+{\left(\varphi \theta \right)}_{lm}\;+{\left(\tau \theta \right)}_{km}+{\left(\tau \varphi \theta \right)}_{klm}+e_{hijklmn}$$
where $y_{hijklmn}$ is the observation of flower *n* of sample type *j* from the *k*th genotype treated with solution *l* and grown in *h*th row and *i*th column measured at day *m* after flowering (DAF), μ is the intercept, $r_{hm}$ and $c_{im}$ are the random effects of the *h*th row and with *i*th column from the Latin square design at DAF *m*, $p_{him}$ is the plant effect of the plant grown in the *h*th row, *i*th column at DAF *m*, $s_{j}$ is a fixed effect for the *j*th sample type, $\tau_{k}$, $\varphi_{l}$ and $\theta_{m}$ are the fixed main effects of genotype *k*, solution *l* and DAF *m*, respectively, ${\left(\tau \varphi \right)}_{kl}$, ${\left(\varphi \theta \right)}_{lm}$, ${\left(\tau \theta \right)}_{km}$, and ${\left(\tau \varphi \theta \right)}_{klm}$ are the two- and three-way interaction effects of the corresponding factors involved, and $e_{hijklmn}$ is the error of $y_{hijklmn}$. Note that observations were weighted either with the number of flowers included or with its inverse depending on whether a mean value (PV) was analysed or the sum across flowers (TPC). Furthermore, as data were repeatedly taken from plants, a first-order autoregressive variance–covariance structure with heterogeneous variances across DAF was assumed. Residuals from this model were checked graphically for homogeneous variance (despite the heterogeneity already modelled) and normal distribution. To fulfil these pre-requirements, PV and TPC were logit and logarithmically transformed, respectively. In this case, means were backtransformed for the purpose of presentation only. Standard errors were backtransformed using the delta method. Additionally, in case of finding differences via global F test, a Tukey test was performed with α = 0.05. For genotype and solution main effects, global F test and Tukey test resulted in the same *p*-value, as only two levels existed. Note that the model fitted an effect for sample type and thus allows that a pooled sample of five flowers had a different expected value compared to single-flower samples.

## 4. Conclusions

Treatment of cannabis plants with either STS or GA3 is known to induce male flower development in female dominant plants. Within this study, the viability and quantity of pollen of two phytocannabinoid-rich genotypes, KANADA (KAN) and A4, were investigated over a period of 8 to 24 DAF using IFC technology. The number of flowers per plant were similar in final sampling dates between treated solutions; however, TPC were significantly different between two solutions at all sampling dates. Treatment with STS led to a significantly higher number of pollen cells and higher viability compared to GA3. Sampling of pollen in a timeframe of 14 to 24 DAF can be recommended for genotype A4 and 8 to 21 DAF for KAN. No significant differences in TPC were found between the tested genotypes; however, further studies including more genotypes are recommended to confirm these results. Moreover, GA3 should be further investigated with various concentrations and in combination with STS to test the efficiency of GA3 on TPC and PV. Furthermore, the study revealed that the number of flowers per plant might not be a suitable single criterion for pollen collection in breeding programs, as PV and TPC were not maximized, when the highest number of flowers per plant occurred. IFC technology has the potential to identify PV and TPC of male flowers in female dominant plants to further assist breeding strategies in medical cannabis.

## Figures and Tables

**Figure 1 plants-10-02739-f001:**
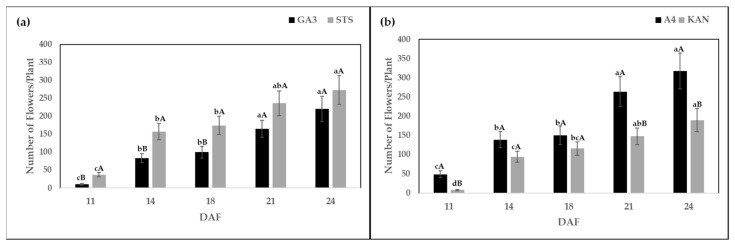
(**a**) Number of flowers per plant after the application of gibberellic acid (GA3) and silver thiosulfate solution (STS) and (**b**) for the tested two genotypes KAN and A4, harvested at 11, 14, 18, 21, and 24 days after flowering (DAF). Means followed by at least one identical lower-case letter did not differ significantly, between DAF (**a**) within a chemical solution and (**b**) within a genotype. Means followed by at least one identical uppercase letter are not significantly different for (**a**) chemical solutions and (**b**) genotypes within a DAF as indicated by Tukey’s test (α = 0.05).

**Figure 2 plants-10-02739-f002:**
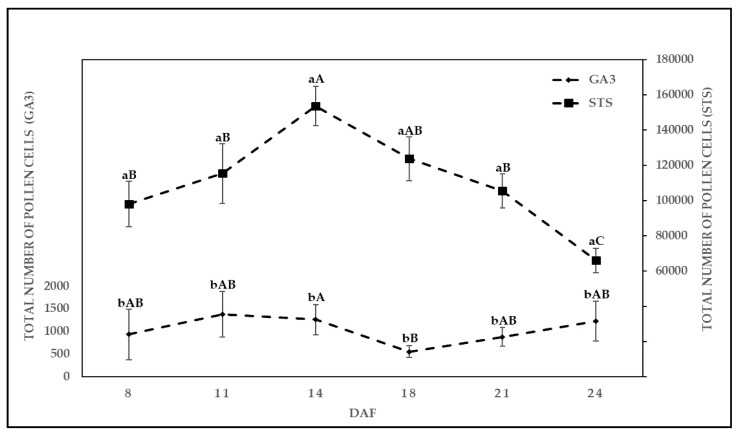
Mean total number of pollen cells (TPC) produced over various days after flowering (DAF) treated with two chemical solutions, gibberellic acid (GA3) and silver thiosulfate solution (STS). Means covered with at least one identical lowercase letter did not differ significantly between two solutions at each date and at α = 0.05. Means covered with at least one identical uppercase letter did not differ significantly at α = 0.05, between DAF as indicated by Tukey test.

**Figure 3 plants-10-02739-f003:**
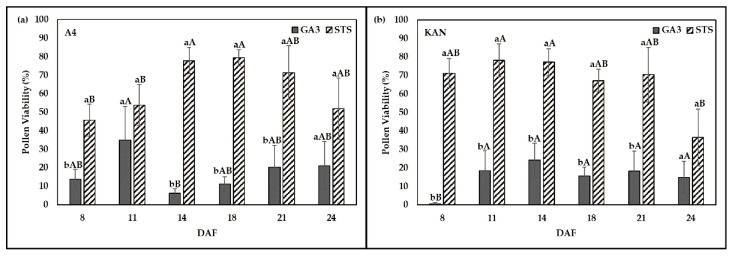
Pollen viability (%) of two genotypes, (**a**) A4 and (**b**) KAN. Both genotypes were treated with two chemical solutions, silver thiosulfate solution (STS) and gibberellic acid (GA3). Pollen was harvested over 24 days after flowering (DAF). Means covered with at least one identical lowercase letter did not differ significantly at α = 0.05 between solutions. Means covered with at least one identical uppercase letter did not differ significantly at α = 0.05, between DAF as indicated by the Tukey test.

**Figure 4 plants-10-02739-f004:**
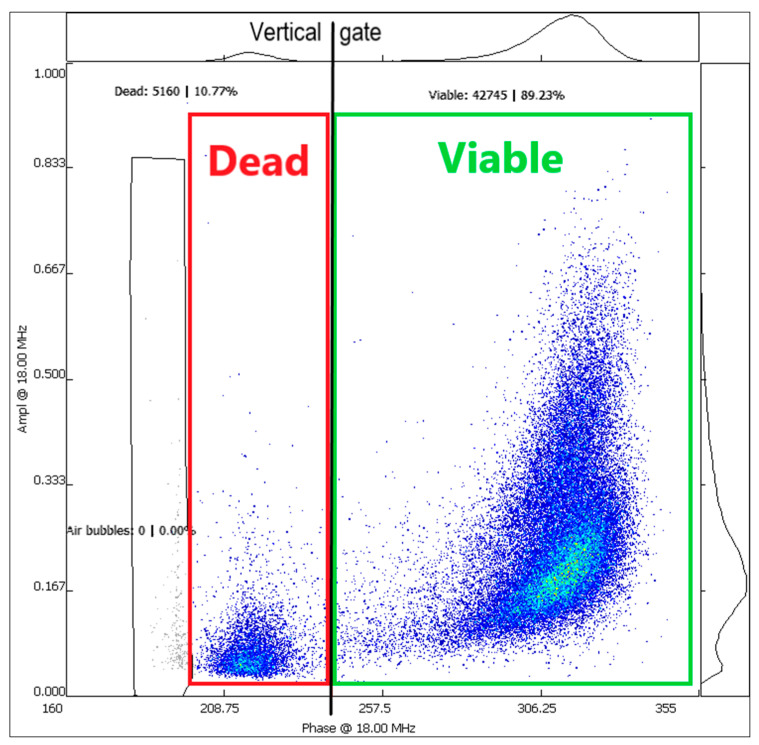
Display of the results in AmphaSoft 2.0 at 18 MHz. Each dot in the scatter plot corresponds to one measured pollen grain. Approximately 50,000 pollen grains in total were measured. Dead pollen grains (red square) appear on the left side of the plot, which can be separated by a gate (black vertical line) from the viable pollen grains (green square).

**Table 1 plants-10-02739-t001:** ANOVA tables for the traits number of flowers per plant, total number of pollen cells (TPC), and pollen viability (PV). *p*-values corresponds to global F tests for differences between the genotypes, solutions, days after flowering (DAF), or their interactions.

Source of Variation	Number of Flowers/Plant	TPC	PV
Genotype	<0.0001	0.2612	0.5753
Solution	<0.0001	<0.0001	<0.0001
DAF	<0.0001	0.0657	0.0194
Genotype × Solution	0.6967	0.3260	0.2789
Genotype × DAF	0.0003	0.1785	0.2582
Solution × DAF	0.0443	0.0266	0.0711
Genotype × Solution × DAF	0.0708	0.1666	0.0183

## Data Availability

Not applicable.

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
