# Peer review of "Potential of Impedance Flow Cytometry to Assess the Viability and Quantity of Cannabis sativa L. Pollen"

_plants, 2021, doi:10.3390/plants10122739_

Round 1

Reviewer 1 Report

The authors have revised the manuscript satisfactory; the manuscript is not ready for publication.

In reference to my comment to the previous submission:

  1. Scope of the study: Previous comment: This submission present a very small data set, which present results for technical testing of a method well used for pollen evaluation in many other plants. It therefore reads like a preliminary testing of a routinely used methodology. The data presented is therefore by itself insufficient for publication in a manuscript. A proper research question is not presented and no such research question is investigated.  Although 2 chemical treatments are used on the plants, these are not discussed or evaluated for their biological effect. Rather, they  are used here to evaluate if the routine technique for pollen viability evaluation works also for cannabis.

My response: The authors have included additional; data in the ms. The scope of the study is now satisfactory.

  1. Conceptual background: The paragraph in lines 44-54 , as well as the abstract and throughout the manuscript include several incorrect statements. For example- most of the medical cannabis market is not based on low THC lines. The conceptual background of the manuscript is therefore lacking.

My response: The authors have sufficiently revised the ms. And put it in the correct framework.

  1. Background: Most of the introduction is only indirectly related to the research question at hand. The research topic is introduced only at the last lines of the introduction, and the reader is left with questions concerning the reason for some of the measurements conducted. Also, the title is not in accord with some of the goals of the study as are defined at the end of the introduction.

My response: The introduction was revised satisfactory.

  1. Reference to published studies: The introduction includes inaccuracies of reference presentation. For example, in section 55-66, studies with plants other than cannabis are mentioned when discussing cannabis vegetative reproduction. The writing should be revised for accuracy and clarity. And the authors are not familiar with some important publications  in their studied area. For example, the last part of the above mentioned paragraph details a phenomenon with reference to manuscripts to this effect that are available for cannabis in the scientific literature.

My response: The text was revised satisfactory to include also appropriate references.

  1. Writing: the English is awkward at some places and should be checked and corrected. One example: in the abstract: "was used the first time with" should be "was used for the first time with".

My response: The English was improved.

  1. Cultivation conditions: Greenhouse conditions are not control environmental conditions. The statement in the abstract and the M&M section are incorrect.

My response: The abstract ans the M&M were revised following this comment. The revisions are OK. .

  1. The statement is line 41-43 is not backed up by scientific references.

My response: The changes conducted to this part of the manuscript are satisfactory.

Reviewer 2 Report

The manuscript is prepared according to Reviewer reamarks from the first round of evaluation. 

This manuscript is a resubmission of an earlier submission. The following is a list of the peer review reports and author responses from that submission.

Round 1

Reviewer 1 Report

The manuscript “Assessing the Viability and Quantity of Cannabis sativa L. Pollen using Impedance Flow Cytometry” focuses for the first time on the application of IFCM in breeding studies in cannabis. The research is novel and of both practical and scientific importance.

The article can be accepted for publication after making some corrections to the text, mentioned below and in the corrected manuscript.

The English form is generally understandable, nonetheless, some grammar, punctuation, and style errors or inconsistencies can be found. Therefore, I suggest showing the text to a native speaker.

MDPI uses the oxford comma (serial comma).

Keywords should be arranged alphabetically. Moreover, please do not repeat keywords from the title.

Names of genera should be written with an upper case letter. On the other hand, if you are referring to a common name, then it should not be in italics (as in most “cannabis” cases in the text).

Throughout the manuscript, the authors are mixing two different terms: variety (wild form) and cultivar (man-made).

All abbreviations must be explained when first mentioned.

Please, provide the producers of key chemicals used in the study.

Some unclear statements in the Materials and methods, Results and discussion, and Conclusion sections require better explanation or style improvement (see the corrected manuscript). For example, how many plants were grown in the greenhouse? What AC was used in the study (line 279)?

Please, provide full botanical names of species when mentioned for the first time (line 285).

Was there any non-treated control included?

In the Reference list, please, follow the formatting style of MDPI.

For more specific comments, please see the corrected manuscript.

After incorporating all the necessary changes, the manuscript can be accepted for publication.

Reviewer 2 Report

This is the interesting study providing important highlights about Viability and Quantity of Cannabis sativa L. Pollen using Impedance Flow Cytometry.

The study updates us with information about the impact of two chemical solutions, silver thiosulfate solution (STS), and gibberellic acid (GA3), in regard to pollen viability (PV) and total number of pollen cells (TPC) of two phyto-cannabinoid-rich (PCR) cannabis genotypes, KANADA (KAN) and A4.

The readability of the manuscript is fluent and understandable. There is one general concern which should be addressed before accepting for publication:

  1. The study is of a high applicative value and the MS is well-written. Since the extend of the study/experiment is too narrow/poor using one relatively simple and well-known application of flow cytometry it is not enough to publish in the Plants journal. Therefore I would suggest perform/include additional analysis using genetic data to complete the study. There is also a lack of standards included in the study.

Reviewer 3 Report

  1. Scope of the study: This submission present a very small data set, which present results for technical testing of a method well used for pollen evaluation in many other plants. It therefore reads like a preliminary testing of a routinely used methodology. The data presented is therefore by itself insufficient for publication in a manuscript. A proper research question is not presented and no such research question is investigated.  

Although 2 chemical treatments are used on the plants, these are not discussed or evaluated for their biological effect. Rather, they  are used here to evaluate if the routine technique for pollen viability evaluation works also for cannabis.

  1. Conceptual background: The paragraph in lines 44-54 , as well as the abstract and throughout the manuscript include several incorrect statements. For example- most of the medical cannabis market is not based on low THC lines. The conceptual background of the manuscript is therefore lacking.
  2. Background: Most of the introduction is only indirectly related to the research question at hand. The research topic is introduced only at the last lines of the introduction, and the reader is left with questions concerning the reason for some of the measurements conducted. Also, the title is not in accord with some of the goals of the study as are defined at the end of the introduction.
  3. Reference to published studies: The introduction includes inaccuracies of reference presentation. For example, in section 55-66, studies with plants other than cannabis are mentioned when discussing cannabis vegetative reproduction. The writing should be revised for accuracy and clarity. And the authors are not familiar with some important publications  in their studied area. For example, the last part of the above mentioned paragraph details a phenomenon with reference to manuscripts to this effect that are available for cannabis in the scientific literature.
  4. Writing: the English is awkward at some places and should be checked and corrected. One example: in the abstract: "was used the first time with" should be "was used for the first time with".
  5. Cultivation conditions: Greenhouse conditions are not control environmental conditions. The statement in the abstract and the M&M section are incorrect.
  6. The statement is line 41-43 is not backed up by scientific references.